# Specifically Regional Cerebral Hypoperfusion in a Case of Highly Suspected Sporadic Creutzfeldt-Jakob Disease on ^99m^Tc-ECD SPECT/CT with Easy Z-Score Imaging System Analysis

**DOI:** 10.3390/diagnostics12020437

**Published:** 2022-02-08

**Authors:** Shun-Chieh Chen, Daniel Hueng-Yuan Shen, Hung-Yen Chan, Ming-Hui Yang, Hung-Pin Chan

**Affiliations:** 1Department of Nuclear Medicine, Kaohsiung Veterans General Hospital, Kaohsiung 813414, Taiwan; schieh1028@gmail.com (S.-C.C.); shen8484@gmail.com (D.H.-Y.S.); hongyenchan0407@yahoo.com.tw (H.-Y.C.); 2Graduate Institute of Science and Technology Law, National Kaohsiung University of Science and Technology, Kaohsiung 80778, Taiwan; 3Department of Medical Education and Research, Kaohsiung Veterans General Hospital, Kaohsiung 813414, Taiwan; mhyang@vghks.gov.tw

**Keywords:** ^99m^Tc-ECD brain SPECT, cerebral hypoperfusion, Creutzfeldt-Jakob Disease, MRI, SPECT/CT

## Abstract

Sporadic Creutzfeldt-Jakob disease (sCJD) is a rapidly-progressive dementing illness, the challenge of diagnosis during life. We presented a 78-year-old woman reported stupor, right facial palsy, and fluctuations in consciousness. ^99m^Tc-ECD brain SPECT/CT with eZIS analysis revealed significant decreased regional cerebral blood flow mainly in specific regions of Alzheimer’s disease as the published article reported with involving frontal region. Brain DWI MRI increased signal intensities corresponding to similar location of ^99m^Tc-ECD brain SPECT/CT. In this case, we reported the pattern of decreased rCBF may correlate to rapidly progressive dementia and associated neurodegenerative symptoms of the highly suspected sCJD patient.

**Figure 1 diagnostics-12-00437-f001:**
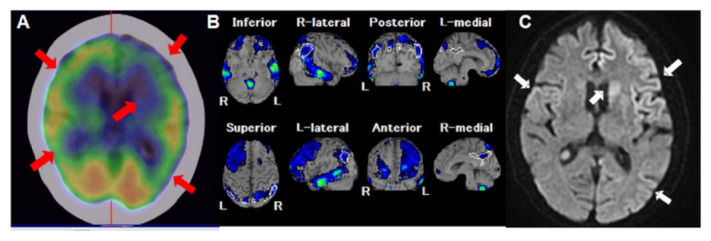
A 78-year-old woman was admitted because of intermittent stupor, right facial palsy, fluctuations in consciousness and bizarre behavior; neurological examination revealed akinetic mutism, startle reflexes, and limbs rigidity. Blood and cerebrospinal fluid (CSF) tests and brain CT was arranged 3 days later after admission that yielded normal finding. After 2 weeks, due to complex partial seizure attack and rapidly progressive dementia, ^99m^Tc-ethyl cysteinate dimer (ECD) single-photon emission computerized tomography/computerized tomography (SPECT/CT) was arranged and it showed hypoperfusion defects in bilateral frontal areas, left caudate and bilateral parietal regions (**A**, red arrows). Easy Z-score imaging system (eZIS) analysis of ^99m^Tc-ECD SPECT imaging revealed significant decreased regional cerebral blood flow (rCBF) in bilateral frontal areas (left > right), bilateral temporal areas, posterior cingulated cortex extending to the precuneus and bilateral parietal cortex (**B**, green and blue regions). After 9 days of ^99m^Tc-ECD SPECT evaluation, brain diffusion-weighted image MRI arranged that showed increased signal intensities over left caudate, bilateral fronto-temporal, and left parietal regions (**C**, white arrows). Due to above findings, CJD was impressed to this patient and she underwent repeat CSF examination. The examination detected positive 14-3-3 protein. However, she expired 16 weeks later due to downhill of clinical outcome. CJD occur sporadically in about 85% of cases, about 10% are inherited, <1% are iatrogenic, and <1% are variant [1]. In published articles, sCJD is a rapidly progressive dementing illness and associated with EEG/neuroimaging abnormalities, or CSF protein changes [2,3,4,5].^99m^Tc-ECD brain SPECT with eZIS analysis for evaluation of early-diagnosis of neurodegenerative diseases was reported by specifically regional cerebral perfusion location [6]. According to this, it mentioned decreased rCBF in the posterior cingulate cortex, precuneus and parietal cortex mainly in Alzheimer’s disease; however, frontal gyrus and insula mainly in frontotemporal dementia. Our patient presented decreased rCBF regions not only specific region of Alzheimer’s disease as prior article reported, but also involving frontal region. In this case, we presented the highly suspected sCJD patient who revealed rapidly progressive dementia with decreased specific rCBF regions in ^99m^Tc-ECD brain SPECT and increased signal intensities of similar locations in brain DWI MRI. The pattern of decreased rCBF may correlate to rapidly progressive dementia and associated neurodegenerative symptoms of the highly suspected sCJD patient.

## Data Availability

Data related to this study cannot be sent to the outside due to information security policies in the hospital.

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
