# Peer review of "Specifically Regional Cerebral Hypoperfusion in a Case of Highly Suspected Sporadic Creutzfeldt-Jakob Disease on ^99m^Tc-ECD SPECT/CT with Easy Z-Score Imaging System Analysis"

_diagnostics, 2022, doi:10.3390/diagnostics12020437_

Round 1
Reviewer 1 Report
This is an interesting imaging finding, showing some correlation of blood flow with cortical ribboning in a patient with CJD. But the written description of the findings is very poor and will need significant revision.
The English will need a lot of improving. Almost every sentence is incorrect, and sometimes the meaning is unclear. They say “significant decreased regional cerebral blood flow mainly in specific regions of Alzheimer’s disease” – this doesn’t make sense. If those regions are specific for Alzheimer’s disease, how do the conclude they are from prion disease, not co-existing Alzheimer disease (which is not unrealistic in a 78yo person). Then they say it involves the frontal region. But that is not the typical region for Alzheimer (which is more parietal) so I don’t understand what they are saying.
They then say the decreased rCBF may correlate to associated neurogenerative symptoms. The symptoms listed are stupor, right facial palsy (but then don’t specify whether this was facial nerve palsy or upper motor neuron facial droop – which have very different localizations), bizarre behaviour, and akinetic mutism, startle reflexes and rigidity. How do they think each of these symptoms correlates with the imaging findings? That is an impossible direct correlation to make and I suggest they remove that statement or else provide some rationale behind the symptoms and locations they are talking about.
It sounds like the patient presented in stupor with akinetic mutism? But then survived 16 more weeks? That is surprising. Usually once they reach akinetic mutism, patients succumb more quickly, unless they are kept alive artificially. A better chronological presentation of the clinical course will help clarify this. It is important because there is a wide range of prion presentations, so any clinical information is helpful (for example, how far along were her clinical symptoms at the time the SPECT and MRI were done).
Was a SPECT really done before MRI? Usually we would default to MRI first…
The legend should be separate from the rest of the text. Right now there is only an abstract, then one long legend. The legend needs to be separated so it only describes the image, then the rest of the text should be the case report.
In the image itself, the description of images are more precisely described as left caudate, not corpus striatum.
Initially they said blood and CSF and CT were “negative”. It would be better stated as “normal”. They should also state what was sent for testing, as there is a long list of options. Was she screened for paraneoplastic conditions for example?
Later in the text they say repeat CSF was positive for 14-3-3, so presumably the first CSF was not sent for that, but they don’t specify this.
Was the case confirmed by autopsy? If not, the authors have to be careful to state this is PROBABLE CJD, not definite. I agree that the limited case information provided, plus the MRI, make CJD very likely. But we cannot conclude that there isn’t co-existing Alzheimer pathology that could give rise, in part, to the SPECT changes. They need to introduce this possibility in the discussion.
Overall it is indeed an interesting image, but it has to be framed in a coherent and interesting way.
Author Response
Dear reviewer:
Please see the attached file. And sorry for a little bit delayed that I suferred from uneven vertigo and now under medicatin controlled.
Dr Chan

Reviewer 2 Report
It is an interesting finding. I would like to suggest following edits.
- Need some English language editing.
- Please include duration of symptoms.
- Please include the timeline of tests, i.e., after how many hours or days of presentation were the tests done.
- Did the diffusion weighted images correlate with ADC?
- Was there any repeat imaging to see if this decrease in cerebral blood flow was progressive or just momentary?
Author Response

(The authors gave the same response as above.)
